# Design, Synthesis, Kinetic Analysis and Pharmacophore-Directed Discovery of 3-Ethylaniline Hybrid Imino-Thiazolidinone as Potential Inhibitor of Carbonic Anhydrase II: An Emerging Biological Target for Treatment of Cancer

**DOI:** 10.3390/biom12111696

**Published:** 2022-11-16

**Authors:** Atteeque Ahmed, Mubashir Aziz, Syeda Abida Ejaz, Pervaiz Ali Channar, Aamer Saeed, Seema Zargar, Tanveer A. Wani, Asad Hamad, Qamar Abbas, Hussain Raza, Song Ja Kim

**Affiliations:** 1Department of Chemistry, Quaid-I-Azam University, Islamabad 45320, Pakistan; 2Department of Pharmaceutical Chemistry, Faculty of Pharmacy, The Islamia University of Bahawalpur, Bahawalpur 63100, Pakistan; 3Department of Basic Sciences and Humanities, Faculty of Information Science and Humanities, Dawood University of Engineering and Technology, Karachi 74800, Pakistan; 4Department of Biochemistry, College of Science, King Saud University, P.O. Box 22452, Riyadh 11451, Saudi Arabia; 5Department of Pharmaceutical Chemistry, College of Pharmacy, King Saud University, P.O. Box 2457, Riyadh 11451, Saudi Arabia; 6Faculty of Pharmacy, Grand Asian University Sialkot, Sialkot 51310, Pakistan; 7Department of Biology, College of Science, University of Bahrain, Sakhir 32038, Bahrain; 8College of Natural Sciences, Department of Biological Sciences, Kongju National University, Gongju 32588, Republic of Korea

**Keywords:** 3-ethylaniline hybrid imino-thiazolidinones, carbonic anhydrase II, DFT studies, molecular docking, molecular dynamic simulation, drug likeness scores

## Abstract

Carbonic anhydrases (CA), having Zn^2+^ metal atoms, are responsible for the catalysis of CO_2_ and water to bicarbonate and protons. Any abnormality in the functioning of these enzymes may lead to morbidities such as glaucoma and different types of cancers including brain, renal and pancreatic carcinomas. To cope with the lack of presence of a promising therapeutic agent against these cancers, searching for an efficient and suitable carbonic anhydrase inhibitor is crucial. In the current study, ten novel 3-ethylaniline hybrid imino-thiazolidinones were synthesized and characterized by FTIR, NMR (^1^H, ^13^C), and mass spectrometry. Synthesis was carried out by diethyl but-2-ynedioate cyclization and different acyl thiourea substitutions of 3-ethyl amine. The CA (II) enzyme inhibition profile for all synthesized derivatives was determined. It was observed that compound **6e** demonstrated highest inhibition of CA-II with an IC_50_ value of 1.545 ± 0.016 µM. In order to explore the pharmacophoric properties and develop structure activity relationship, in silico screening was performed. In silico investigations included density functional theory (DFT) studies, pharmacophore-guided model development, molecular docking, molecular dynamic (MD) simulations, and prediction of drug likeness scores. DFT investigations provided insight into the electronic characteristics of compounds, while molecular docking determined the binding orientation of derivatives within the CA-II active site. Compounds **6a**, **6e**, and **6g** had a reactive profile and generated stable protein-ligand interactions with respective docking scores of −6.12, −6.99, and −6.76 kcal/mol. MD simulations were used to evaluate the stability of the top-ranked complex. In addition, pharmacophore-guided modeling demonstrated that compound **6e** produced the best pharmacophore model (HHAAARR) compared to standard brinzolamide. In vitro and in silico investigations anticipated that compound **6e** would be an inhibitor of carbonic anhydrase II with high efficacy. Compound **6e** may serve as a potential lead for future synthesis that can be investigated at the molecular level, and additional in vivo studies are strongly encouraged.

## 1. Introduction

Imino-thiazolidinones with sulphur and nitrogen atoms [1] have remained in focus due to their wide range of pharmacological [2] and biological activities [3] such as anticancer [4], antifungal [5], anti-inflammatory [6], antimicrobial [7], antidiabetic [8,9], anti-HIV [10], anti-schistosomal and utility for the treatment of various skin diseases [11,12]. Based upon the nature of these substituents, variety of medicinally important compounds can be synthesized. Major examples include necroptosis inhibitors [13], selective GSK-3β inhibitors [14], anticancer agents [15], anti-inflammatory agents [16], potent antiproliferative agents [17], and matrix metalloproteinase-3 inhibitors [18]. Versatile activities of the imino-thiazolidinone scaffold are shown in Figure 1, which were based on the pendent groups R_1_, R_2_, and R_3_ involved.

The ubiquitous distribution and catalytic activity of carbonic anhydrase in various physiological processes made them a selective target for development of new inhibitors. Over the last few years, serious attempts were made to explore the role of carbonic anhydrase (CA) in tumor progression, either as biomarker or a tumor-associated protein [19]. Recently, a library of convincing evidence suggested that the excess of CA isozyme in some cancers provide acidic medium in extracellular matrix, which in turn encourage the growth and metastasis of tumor [20,21]. The CAI and CAII expression have been detected in various tumor cells and cell lines, but it has been tough to get clear picture of the relationship between malignant and normal cells. Among the CAs, isoform CAII is highly active and has a broad cytosolic distribution. Intensive expression for CAII was detected in neoplastic and normal ductal cell of pancreatic exocrine tissue. Normal colonic epithelium shows both CAI and CAII. It was found that there is negative correlation between the expression of CAI and CAII in colorectal malignant mucosa and the proliferation of cells. Aforementioned evidence has made CAs an attractive target for the treatment of certain cancers [22,23,24]. For the extension of applicability of imino-thiazolidinones to carbonic anhydrase II inhibition, we have synthesized novel Ethyl (Z)-2-((Z)-3-((3s, 5s, 7s)-adamantan-1-yl)-2-((2-methylbenzoyl) imino)-4-oxothiazolidin-5-ylidene) acetate (**6f**) and its derivatives (**6a**–**j**). Inhibitory activities were tested by using the standard protocols as mentioned in the experimental section. Moreover, molecular docking studies and DFT studies were performed to obtain insight into the binding mechanism and electronic structures.

## 2. Results and Discussion

### 2.1. Synthesis and Characterization

The synthesis of imino-thiazolidinone derivatives were carried out according to a Figure 1 [25] synthetic approach. Using DCM as a solvent, substituted benzoic acids were transformed to matching acid chlorides. The reaction between acid chloride and potassium thiocyanate produced isothiocyanate in acetone. In situ reaction of the latter with an equal amount of 3-ethylaniline (4) produced acyl thioureas (**5a**–**j**). The reaction of ethyl 4-ethoxypent-4-en-2-ynoate with (**5a**–**j**) in dry ethanol at room temperature produced the required compounds (**6a**–**j**) with outstanding yields and high purity [26]. The 2D structures of novel imino-thiazolidinone derivatives are shown in (Figure 2).

The FTIR spectral data of newly synthesized imino-thiazolidinones (**6a**–**j**) showed C-H aliphatic, asymmetric, sp^3^, C-H aliphatic, symmetric, sp^3^ stretching at 3100–2850 cm^−1^, whilst carbonyl of acyl, ester, and thiazolidine were shown at 1738–1731, 1702–1692, and 1651–1637 cm^−1^, respectively. In ^1^H-NMR spectrum, singlets integrating ^1^H each appeared in the range of 7.11–7.06 ppm for the α,β-unsaturated alkene proton and aromatic protons 7.97–7.14, and in addition to the two quartets, one for ester methylene protons integrating 2H in the range of 4.38–4.38 ppm and other for aromatic ethyl group methylene in the range of 2.77–2.75 ppm. Furthermore, two adjacent triplets with integration 6H were observed in the range of 1.42–1.29 ppm for two methyl groups of ethyl part in the molecule.

The ^13^C NMR spectrum (provided in Appendix A) further confirmed the assigned structure by displaying the characteristic signals for acyl carbonyl carbon in the range of 178–165 ppm and three signals of carbon for carbonyl of ester, thiazolidine, and azomethane carbon in the range of 165–155 ppm indicated the synthesis of the desired molecular design. Additionally, two ethyl part carbon signals in the aliphatic, alpha carbonyl and benzyl region indicate the synthesis of molecule. The IR, ^13^C NMR, and ^1^HNMR spectra with comprehensive characterization data are provided in the Appendix A. The 3-ethylphenyl was selected as the lipophilic tuning part of the pharmacophore and to develop hydrophobic interactions of the ligand with the protein active site.

### 2.2. Carbonic Anhydrase Activity

In the present work, (Z)-ethyl 2-((Z)-2-benzamido-3-(4-ethylphenyl)-4-oxothiazolidin-5-ylidene) acetate **(6a**–**j**) derivatives were synthesized with the aim of inhibiting the carbonic anhydrase II enzyme. In the newly synthesized **6a**–**j** derivatives, methyl and halogen groups were substituted at the phenyl ring.

The substituted functional group had a substantial effect on the inhibitory potential of all derivatives against CA-II activity. Specifically, substitution of methyl and chlorine atom at benzene ring had significant impact on the inhibitory potential of the compound. Compound **6a** had a substitution of methyl group at the *ortho* position of the benzene ring, which demonstrated IC_50_ 1.665 ± 0.095 µM. Similarly, compound **6e** possessed substitution of methyl group at the meta position, which acted as an electron-donating group and improved the efficacy of the compound against CA-II. The IC_50_ value of the compound **6e** was 1.545 ± 0.016 µM, which is best among all derivatives. Furthermore, substitution of the halogen atom also possessed a pivotal role in establishing the efficacy of derivatives. In particular, compound **6g** possessed a substitution of chlorine atoms at the aromatic ring, which improved its inhibitory potential. The present work employed acetazolamide as standard inhibitor, which exhibited strong inhibitory potential against CA-II (IC_50_ 1.089 ± 0.063 µM). The standard inhibitor possesses a thiadiazole ring and an acetamide moiety in its structure, which is responsible for its activity. On the other hand, synthesized derivatives can be divided into three essential pharmacophore parts: (1) central oxothiazolidine, (2) benzamide, and (3) acetate moiety. These pharmacophores are essential for anti-carbonic anhydrase activity. Specifically, the presence of sulfonamide and thiadiazole rings in acetazolamide resulted in substantial inhibitory potential. Similarly, substitution of the benzene ring at the central oxothiazolidine ring of compounds **6a** and **6e** resulted in a significant improvement in inhibitory potential, which was comparable to the standard inhibitor. These findings are suggestive of the strong inhibitory potential of synthesized derivatives against CA-II. The inhibitory potential of synthesized derivatives along with a standard drug is provided in Table 1.

### 2.3. Structure–Activity Relationship

All the synthesized derivatives were substituted different functional groups including: CH_3_, Cl, F, and Br at the phenyl ring. The derivative **6f** was kept unsubstituted. The structure activity relationship of all these compounds was compared with this derivative **(6f)**. Among all the derivatives, compounds **6a, 6e**, and **6g** were considered the best inhibitors and they showed almost equipotent inhibition potential against carbonic anhydrase, i.e., 1.665 ± 0.095, 1.545 ± 0.016, and 1.784 ± 0.095, respectively. Although, a slight difference in the inhibitory values was found, which was negligible. When the structure activity relationship of these derivatives was compared with **6f** it was found that the induction of the CH_3_ group at the meta position resulted in the improved inhibitory activity of the compound **6e**, making it the most effective derivative among all. Interestingly, the activity was found to be reduced when the CH_3_ group shifted on the ortho position as in case of compound **6a**. Another interesting behavior was observed when the structure activity relationship between derivatives **6g**, **6i**, and **6b** was compared, all containing the same Cl substituent. It was found that the ortho and para position did not agree with the inhibitory potential of these derivatives against CA-II, as can be observed in case of **6i** and **6b**. The Cl at the meta position improved the inhibitory potential of the compound. In comparison to the inhibitory values of derivative **6g**, the activity of **6i** and **6b** was found to be reduced approximately 2.9- and 39.9-fold, respectively. Moreover, the meta chloro-substituted derivative **6g** showed higher activity than compounds **6c** having bromo substitution at the ortho position. This decrease in activity can be related to presence of a bulkier electron withdrawing group Br at the ortho position (**6c**), which produced strong electron deficiency at the C center of the aromatic ring. It was observed that the mono substitution of less bulky electron-withdrawing substituents (F, Cl) at the ortho and/or para position does not produce a profound withdrawing effect; hence, compounds **6j, 6i**, and **6h** showed moderate inhibition potential.

The overall activity pattern in relation to the structures of derivatives suggested that meta position was the most favorable site for substitution of electron donating or electron withdrawing groups, while ortho and para positions resulted in the reduction of inhibitory activity.

### 2.4. Kinetic Mechanism of Carbonic Anhydrase Inhibition

Based on the IC_50_ values, the most efficacious carbonic anhydrase inhibitor, compound **6e**, was designated for kinetic investigation. Lineweaver-–Burk plots and a second plot are depicted in (Figure 3A,B). At various concentrations of the inhibitor and substrate, Lineweaver–Burk plots (plot of 1/V vs. 1/[S]) were generated to determine the kind of inhibition. As depicted in Figure 3A, the findings of Lineweaver–Burk plots for compound **6e** indicate that V_max_ remains unchanged without significantly modifying the slopes, whereas K_m_ increases with increasing concentrations and V_max_ remains unchanged with a non-significant difference. This shows that the **6e** chemical inhibits carbonic anhydrase through a competitive mechanism. The EI dissociation constant was shown by the second plot (Figure 3B) of the slope vs. the concentration of **6e**. From the inhibitor concentrations of **6e** vs. the slope, the K_i_ value was determined to be 2.2 µM.

### 2.5. In Silico Studies

#### 2.5.1. Density Functional Theory Studies

##### Optimized Structures

The structural geometries of synthesized derivatives were optimized using B3LYP functional correlation and 6-31G* as a basis set. All structures were optimized to the steepest energy gradient. In addition, frequency calculations were also performed. No imaginary frequency was found that depicted present geometries as authentic local minimums. The optimized structures of potent derivatives are provided in Figure 4, whereas geometries of other compounds are provided in the Appendix A.

##### Frontier Molecular Orbital Analysis

FMO (Frontier Molecular Orbital) calculation has a pivotal role in the estimation of stability and reactivity patterns as these orbital represent the electron occupancy, i.e., HOMO represents the highest occupied molecular orbitals while lowest electron occupation are indicated by LUMO [27]. Frontier molecular orbitals were calculated by using B3LYP/6-31G*. Maximum HOMO delocalization was observed at one of the terminal phenyl ring in all compounds confirming presence of resonance at mentioned location. Only compound **6c** showed maximum electron density at both phenyl rings. Maximum LUMO delocalization was present at the thiazolidine ring and its vicinity. Almost complete HOMO delocalization in **6i** was observed at terminal phenyl ring. Maximum LUMO delocalization was present at the thiazolidine ring and its adjacent hetero atoms. However, an interesting observation was that the electron density was maximized with little or no dispersion to other atoms. FMO orbitals of compound **6a**, **6e**, and **6g** are provided in Figure 5, whereas FMO orbitals of other derivatives are provided in the Appendix A.

##### Local and Global Reactivity Descriptors

The HOMO and LUMO energy values were utilized to calculate different local and global reactivity descriptors. These descriptors are important metrics in determining the reactivity and stability of the compound [28]. The chemical softness and chemical hardness determine the chemical reactivity of the compounds. A compound having good value of chemical softness tends to be a reactive molecule, e.g., compound **6b** exhibited a chemical softness of 7.2, which illustrates its reactive nature. The detailed reactivity descriptors of all derivatives are provided in Table 2.

Other important chemical reactivity parameters are ionization energy, electron affinity, electron donating power, and electrophilicity index. The electrophilicity index revealed information about dynamics, bonding, stability, reactivity, and toxicity of the compound. This property measures the propensity to electron acceptance. According to the electrophilicity data presented in Table 2, compound **6g** exhibited the highest ability of electron intake with an electrophilicity value of 0.103. In contrast, compound **6a** had the lowest propensity to electron acceptance. In addition, ionization energy referred to the ability of the compound to lose its outermost electron, whereas electron affinity is the tendency of the compound to accept/attract an electron. Compound **6g** had the highest potential of ionization with a value of 0.250 eV. In contrast, compound **6b** exhibited strong potential for electron affinity with a value of 0.1102. The detailed value of each derivative is tabulated in Table 3.

#### 2.5.2. E-Pharmacophore-Based Drug Discovery

Brinzolamide (marketed as Azopt^®^) is reported as a selective, non-competitive [29], and specific inhibitor of carbonic anhydrase II (CA-II) [30]. Considering the availability of Brinzolamide in a complex with CA-II, structure-based pharmacophore modeling was performed during the current investigation (PDB ID: 1A42). The identified molecular interactions between CA-II and brinzolamide served as a model for the development of pharmacophoric characteristics. Including hydrogen bond donor, acceptor, aromatic, and hydrophobic interactions, a total of eight characteristics were created. All synthesized compounds were submitted to pharmacophore-based virtual screening against features that were developed.

Initially, a pharmacophore model was built utilizing the pharmacophore query editor of Molecular Operating Environment version 2014.15 [31]. Generated features of the brinzolamide-CA-II complex consisted of three hydrogen bond acceptors, two hydrophobic, one aromatic, and two hydrogen bond donor atoms (AAAHHRD). The ten types of imino-thiazolidinine derivatives (Figure 2) were evaluated against a generated pharmacophore model. It was observed that RMSD values were between 2.7 and 3 angstroms for compounds **6a** and **6e**, which produced the best pharmacophore models. Compound **6e** included two hydrophobic interactions, three hydrogen bond acceptor contacts, and two aromatic interactions (HHAAARR). The **6e** molecule had a docking score of −6.94 kcal/mol, which was superior to other derivatives. The generated features of compound **6e** are illustrated in the Figure 6. Moreover, compound **6e** produced significant interactions with the target protein, including hydrogen bonding and hydrophobic interactions. These identified features reflect the resemblance of synthesized derivatives with standard brinzolamide. These results bolster the inhibitory ability of synthesized derivatives, particularly compound **6e**, against CA-II.

#### 2.5.3. Molecular Docking Studies

The synthesized derivatives were subjected to molecular docking studies. These docking studies are critical in determining the binding orientation of selected compounds inside the active pocket of the targeted protein [32]. Initially, amino acid residues of the active site were selected and grid box were generated at the active site. The active site residues were as follows: TRP5; ASN62; HIS64; ALA65; ASN67; GLN92; HIS94; LEU198; THR199; THR200; PRO202, VAL207; TRP209; ZN1002; HIS96; HIS119; VAL121; PHE131; VAL143; and SER197.

All synthesized derivatives were subjected to molecular docking studies. It was observed that all derivatives exhibited moderate to excellent docking scores. Specifically, compounds **6a**, **6e**, and **6g** exhibited excellent docking scores. These scores and molecular interactions were in accordance with findings of in vitro (IC_50_) activity and a structure-based pharmacophore modelling approach. The docking scores and binding interactions of compound **6a**, **6e**, and **6g** are presented in Table 4.

The docked conformation of compound **6a** exhibited potential bonding and nonbonding interactions. It was observed that compound **6a** engaged HIS94, in hydrogen bond with the oxygen atom of benzaldehyde with bond length of 3.21 angstrom. A single hydrogen bond was observed between the oxygen atom of the oxothiazolidine ring and THR200. The bond length of 2.85 angstroms demonstrate its high strength. The residues VAL121, and GLN92 were involved in alkyl interactions with the terminal methyl group substituted at the thiazolidine ring. The amino acid residues LEU198 and PRO201 formed pi–alkyl interactions with the thiazolidine ring and benzene rings, respectively. Pi–Sigma interactions were observed between the methyl group attached to the benzene ring and PHE131. Similarly, the benzene ring was also involved in pi-stacked interaction with PHE131. Overall interactions in **6a** were found favorable for the binding of **6a** with CA-II.

Docked conformations of compound **6e** and **6g** also exhibited stronger molecular interactions. Both compounds engaged important amino acid residues in potential interactions including hydrogen bonding and hydrophobic interactions. An interesting pi–cation interaction was observed between the zinc metal and benzene ring. Compound **6e** engaged GLN92, ASN67 and ASN62 in carbon hydrogen bonding with a bond length of 3.29, 2.97 and 3.17 angstroms respectively. Other amino acid residues including ASN244, THR200, TYR7, HIS64, PRO201, ALA65, TRP5, HIS96, LEU198, PHE131, and VAL143 were involved in hydrophobic interactions. In case of compound **6g**, phenyl ring was substituted with Cl atom and this part was also involved in four hydrophobic interactions including alkyl, pi–alkyl, and aromatic interactions with TYR7, ASP243, LEU240 and VAL242 respectively. However, an additional hydrophobic interactions were observed between GLY6, PHE231, and GLY63 with the ethyl-substituted benzene ring present on oxothiazolidine moiety of compound **6g**. In addition, the docking score of compound **6e** was observed to be −6.99 kcal/mol. The detailed 2D and 3D interactions of the compounds **6e** and **6g** are presented in Figure 7. The detailed analysis and 2D interactions of other derivatives are provided in the Appendix A.

The correlation coefficient was identified using a statistical package of social sciences and found statistically significant as there was a significant positive relationship between in vitro activity and the computational activities of the compound. For example, compound **6a**, **6e**, and **6g** had strong in vitro activities and ultimately high docking scores were obtained from computational work (results provided in the Appendix A).

#### 2.5.4. Molecular Dynamic Simulations

Molecular docking studies are effective for generating the first static protein–ligand complex, but they provide little insight into the stability and structure of the complex. To test the stability and dependability of molecular interactions, molecular dynamics simulations (MD) were performed. Root mean square deviation (RMSD) measures the average variation of atoms relative to the standard frame, whereas root mean square fluctuation is used to evaluate the fluctuation of individual amino acid residues in proteins (RMSF). In addition, SASA (solvent accessible surface area) matrices were utilized to evaluate the compactness and stability of protein in solvent.

The top-ranked CA-II-6e complex conformation was derived from molecular docking and subjected to MD simulations. Throughout the simulated trajectory, both the CA-II protein (purple colored trajectory) and the CA-II-6e complex were seen to display a constant RMSD pattern. Specifically, the RMSD pattern of proteins grew from 1.2 to 2.0 angstroms, an extremely consistent characteristic. Throughout the duration of simulation, the trajectory of protein remained equilibrated and steady. Regarding the CA-II-6e complex (green colored trajectory), the protein–ligand combination displayed minor alterations of up to 20 ns of simulation, which were stabilized after 20 ns. After 25 ns, the RMSD pattern of the CA-II-6e complex remained steady. The average RMSD for the CA-II-6e complex was 2.92 angstroms, which is quite acceptable [33]. Compound **6e** may be a possible CA-II inhibitor with substantial anticancer effects, based on the development of the RMSD pattern and the establishment of significant interactions with active amino acid residues, GLN92, and PHE131. In addition, the RMSD pattern for CA-II-6a and CA-II-6g was also evaluated. Figure 8 depicts the RMSD pattern of the CA-II, CA-II-6e, CA-II-6a, and CA-II-6g complexes, respectively. It can be observed that the brown-colored trajectory in Figure 8 illustrates the stability pattern of the CA-II-6a complex. The complex became stable shortly after the commencement of the MD simulation, i.e., 10ns. The fluctuation of the RMSD was in the order of 0.9 angstroms, which demonstrates that the complex didn’t undergo extensive conformational changes. The average RMSD of the CA-II-6a complex was 2.45 angstroms, which lies within an acceptable range. In contrast, the CA-II-6g complex exhibited a slightly higher RMSD pattern (pale yellow trajectory). Initially, CA-II-6g demonstrated a slightly unstable MD trajectory, which became stable after 20 ns and remained equilibrated for the rest of the simulation time. The significant fluctuations might be due to substantial conformational changes of the ligand inside the active pocket of the CA-II. After 20 ns, promising contacts were made with important amino acid residues of protein, which had stabilized the simulated trajectory. The average RMSD value for CA-II-6g was observed to be 4.9 angstroms, which is slightly above the acceptable limit. These findings assist the results of in vitro data. It can be suggested that compounds **6a** and **6e** could serve as promising inhibitors of CA-II, whereas compound **6g** exhibited slightly less inhibiting potential.

To assess the stability and structural changes of ligands inside the protein’s active pocket, a thorough RMSD analysis was carried out. Compounds **6a**, **6e**, and **6g**’s RMSD patterns were assessed. With RMSD values of under 2.5 angstroms, all three compounds displayed a noticeable stability pattern. Consequently, the brown-colored trajectory in Figure 9 shows how the RMSD for compound **6e** has changed over time. Due to the fact that compound **6e** had seven torsions and eight rotatable bonds in total, it exhibited minor structural alterations. The RMSD pattern was remarkably stable and remained in equilibrium.

Conversely, compound **6a** (blue-colored trajectory) showed minute fluctuations and structural changes that stabilized after 20 nanoseconds. The average RMSD for compound **6a** was 2.4 angstroms. Furthermore, the total number of rotatable bonds for compound **6a** was nine, with eight torsions. In terms of compound **6g**, the RMSD patterns depict the substantial conformational changes that resulted in fluctuations of the order of 1.0 angstrom. Compound **6g** exhibited the highest RMSD value of 2.5 angstroms, which became stable after 30 ns. The number of rotatable bonds was nine in compound **6g**, which might be the reason for the slight fluctuations. However, RMSD patterns remained within an acceptable limit for all three selected compounds. Figure 9 illustrates the stability pattern of compounds **6a**, **6e**, and **6g**.

The average variance of amino acid residues throughout simulations was accounted for by the root mean square fluctuation (RMSF). The RMSF value of CA-II was measured, and a steady profile was seen for amino acid residues. The average RMSF value of CA-II residues was 0.8 angstrom, a rather stable result. The development of the RMSF graph reveals that only the C and N terminal residues fluctuated along the simulated flight; the remaining residues stayed constant. Figure 10 illustrates the RMSF values for the CA-II protein.

The solvent accessible surface area (SASA) is an essential measure for determining the compactness and integrity of targeted proteins. If a protein has a high SASA value, it is exposed to the surrounding surface area, whereas a low SASA value indicates less exposure to the surrounding solvent and hence more stability. In the present work, residue-specific SASA values were determined. It was noticed that residues in the range of 85 to 100 were highly exposed to the surrounding solvent and exhibited pronounced fluctuations. The overall SASA score stayed within the recommended range of 200 to 349 Å^2^. The residue-wise SASA value is illustrated in Figure 11.

The pose clustering approach was utilized to determine stability and eliminate the unlikely poses obtained from molecular docking. Initially, 100 poses were generated through AutoDock Vina, which were ranked on a basis of energy rankings. For MD simulations, the pose with the lowest energy value was chosen. Inclusion of a solvent model and integration of atom movement made MD simulations an expensive approach [34]. Thus, for MD simulations, the top three compounds were selected on the basis of energy rankings and subjected to MD simulation studies for stability evaluation and further elimination of unlikely poses. It was observed that compound **6e** remained stable with minimal deviations of less than 2.5 angstroms, whereas compound **6g** exhibited slightly unstable patterns with a pose deviation of more than 3 angstroms. In addition, compound **6a** also remained stable and floated between two and three angstroms. The pose clustering approach helped further eliminate undesired poses. Figure 12 illustrates the pose conformations of selected derivatives obtained from molecular docking and deviations observed during MD simulations.

#### 2.5.5. MM-GBSA Energy Calculations

An effective method for ascertaining the binding affinities of docked conformations is to use MM-GBSA energy calculations. Although the molecular docking method accurately estimates the molecular interactions between a protein and its ligand, it is ineffective at determining the conformations’ binding affinities. As a result, the MM-GBSA approach provides an accurate estimation of the free energies by accounting for all electrostatic, hydrophilic, and hydrophobic interactions. The binding free energy was calculated using a script-based technique [33]. Free energies from the MM-GBSA technique were found to be more negative and superior to docking scores from molecular docking. Additionally, vdw, coloumb, and H-bond energies all contributed to the identification of compound **6a**, **6e**, and **6g**’s binding affinities. The following chemical equation was employed to determine Gibbs free energy:Δ*G* = Δ*G_SA_* + Δ*G_sol_* + Δ*E_mm_*(1)

The MM-GBSA energies of all three complexes are displayed in Table 5.

#### 2.5.6. In Silico ADMET Predictions

Comprehensive in silico and in vitro findings suggest compound **6e** as a potential lead against CA-II. In order to determine the drug-likeness profile of **6e**, detailed ADMET properties were predicted using deep learning architectures. MolDesigner is an advanced tool used for designing efficacious drugs with neural networks [35]. It utilizes the message-passing neural network (MPNN) for predicting the ADMET profile of a drug-like candidate. In current study, compound **6e** was subjected to in silico ADMET prediction and it was observed that **6e** possessed drug-like properties with an optimal toxicity profile. The predicted clinical toxicity and HIA was 14.24% and 92.64%, respectively. Detailed ADMET properties of compound **6e** are tabulated in Table 6.

## 3. Materials and Methodology

### 3.1. General Information

High grade solvent and chemical reagents were purchased from Sigma-Aldrich. The melting points were determined using an uncorrected digital Gallenkamp (SANYO) model MPD BM 3. Using a Bruker AM-300 spectrophotometer, ^1^H NMR and ^13^C NMR spectra were acquired in CDCl_3_ or acetone-d6 solutions at 300 MHz and 75.4 MHz, respectively. FTIR spectra were collected using an FTS 3000 MX spectrophotometer, mass spectra (EI, 70 eV) with a GC-MS device, and elemental analyses with an LECO-183 CHNS analyzer. Chemical shifts were measured in ppm units, while coupling constants (*J*) were measured in hertz.

### 3.2. Procedure for of Synthesis (***6a–j***)

Under inert conditions, 3.97 mmol of substituted aryl carboxylic acid (1.2 mol eq) was reacted with 4.30 mmol of thionyl chloride (1.3 mol eq) at reflux in dichloromethane. After the reaction was complete, dichloromethane was drained under decreasing pressure. By adding 4.96 mmol of potassium thiocyanate to acetone, the resulting acid chloride was transformed to isothiocyanate. The reaction mixture was heated at 40 degrees Celsius for 1.5 h with 3.31mmol of 4-ethylaniline (1 mole equivalent). The crude product precipitated in pure ice and recrystallized in ethanol. The resultant acyl thioureas (1.27mmol, 1 mol eq) (**5a–j**) were treated with ethyl 4-ethoxypent-4-en-2-ynoate at room temperature (2.54mmol, 2 mol eq). The product precipitated throughout the course of the procedure. The substance was filtered, dehydrated, and then recrystallized in ethanol. Using the same method, more compounds were synthesized (Figure 1)**.**

### 3.3. Carbonic Anhydrase Assay

As mentioned previously, the inhibition of carbonic anhydrase was investigated with adjustments [36,37]. The method relies on the fact that carbonic anhydrase hydrolyzes p-nitrophenyl acetate into yellow-colored p-nitrophenol, which is then quantified spectrophotometrically. Each well contained 120 µL of a 50 mM Tris-sulfate buffer (pH 7.6, containing 0.1 mM ZnCl_2_), 20 µL of an inhibitor, and 20 µL of bovine enzyme (50 U). The materials were properly mixed and incubated for 10 min at 25 degrees Celsius. The substrate p-nitrophenyl acetate was synthesized (6 mM stock using <5% acetonitrile in buffer and used fresh each time) and 40 µL was added per well to achieve a concentration of 0.6 mM per well. The total volume of the reaction was 200 µL. After 30 min of incubation at 25 °C, the absorbance of the reaction mixture was measured at 348 nm using a microplate reader (SpectraMax ABS, San Jose, CA, USA). The standardization process utilized acetazolamide. Each concentration was assessed in three distinct tests. The IC_50_ values were calculated with GraphPad Prism 5.0 [38] and nonlinear regression (GraphPad, San Diego, CA USA).
(2)$$\mathrm{Inhibition}\;(\%)=\left[\left(B-S\right)/B\right]\times 100$$

Here, B and S represent the absorbances of the blank and sample, respectively.

### 3.4. Kinetic Analysis

To investigate the kinetic properties of compound **6e**, identical experiments to those described in previous study [36] were performed. The compound **6e** was selected based on its IC_50_ value at various concentrations (0.00, 0.78, 1.55, and 3.11 µM, respectively), whereas the substrate concentrations (p-nitrophenyl acetate) were varied from (2, 1, 0.5, 0.25 and 0.125 mM). Other experimental settings mirrored those mentioned in the section under “Carbonic anhydrase assay”. The maximum starting velocity was estimated using the first linear section of absorbance up to 10 min at 1 min intervals. To evaluate the kind of enzyme inhibition, Lineweaver–Burk plots depicting the inverse of velocities (1/V) vs. the inverse of substrate concentration (1/[S] mM^−1^) were utilized. The EI dissociation constant K_i_ was calculated using the secondary plot of 1/V versus inhibitor concentrations. Version 11 of Microsoft Excel was used to create graphs.

### 3.5. E-Pharmacophore Based Drug Discovery

The description of protein–ligand interactions with pharmacophores is a well-established strategy in modern drug discovery techniques. Pharmacophores are a collection of electronic and steric characteristics that are required to ensure efficient supramolecular interactions with biological targets and to block or initiate the biological response [39]. In brief, pharmacophores consist of certain pharmacophoric characteristics that explain the significant physicochemical properties of ligands. These physicochemical qualities (features) consist of hydrogen bond acceptors/donors, hydrophobic interactions, cationic and ionic groups, and hydrophobic/aromatic interactions. All of these characteristics contribute to the binding of the investigated ligands to the biological target [40].

In e-pharmacophore drug discovery, both structure-based and ligand-based pharmacophore drug design methods are utilized. The selection of a particular method is mostly determined by the available resources and the nature of the investigation(s) [41]. As the crystallographic structure of CA-II in association with a selective, non-competitive, and specific inhibitor (brinzolamide) is accessible in the protein data bank, we employed a structure-based pharmacophore modelling technique in our current investigation. On the basis of protein–ligand complex molecular interactions, pharmacophoric characteristics were characterized (Figure 13). This method ensures the usage of ligand molecules already involved in establishing substantial interaction with the biological target.

After retrieving crystallographic structure from PDB, pharmacophoric features were generated using Molecular Operating Environment 2014.10. Total 8 features were generated including hydrogen acceptor (Acc), hydrogen donor (Don), hydrophobic (Hyd), and aromatic interactions. Afterward these features were used as a template to characterize the activity of synthesized compounds.

### 3.6. Density Functional Theory (DFTs) Studies

All structural geometries of synthesized compounds were subjected to steepest energy optimization. The geometries of each compound was optimized using DFT/B3LYP functional correlational and 6-31G* as a basis set [42]. DFT analyses provide precise estimates of the electronic density of substances. Estimating electronic density is essential for determining the reactivity and stability of compounds. The frontier molecular orbital analysis was also undertaken since HOMO/LUMO energy gaps have a major impact on chemical reactivity [43]. Also determined were the chemical hardness and softness of the compounds. In a multi-core system, Gaussian 09W [44] software was used to conduct quantum mechanics investigation, and GaussView 6 [45] was utilized to inspect the optimized structures.

### 3.7. Molecular Docking Studies

Molecular docking plays a pivotal role in in silico modelling and is proving as an advanced tool in drug design [46]. Molecular docking is performed to assess ligand protein interaction in terms of docking score and binding orientation [47]. The reliability and reproducibility of search algorithms and scoring functions were assessed by validation of docking protocol. The targeted protein was redocked with a co-crystal ligand and reproducibility of docked poses were assessed. The docking protocol was considered validated when root mean square deviation (RMSD) of native pose and regenerated pose was less than 2 Å [48].

The current study has utilized AutoDock Vina [49] for prediction of potential interactions between protein and selected compounds. AutoDock Vina is an efficient suite for molecular modelling with enhanced sampling power, accuracy, and time-efficient manner. For molecular modelling, crystal structure of human carbonic anhydrase II (CA-II) complexed with Brinzolamide was downloaded from the protein data bank (PDB id: 1A42; resolution: 2.24 Å). Initial structure preparation of CA-II was carried out using MGL tools [49]. Protein preparation and preprocess wizard included removal of water molecules and hetero atoms except Zn metal ion. Afterward, protonation was carried out and Gasteiger charges were incorporated. Protein was also assessed for any missing residues that were corrected using proposed approach.

After completion of protein preparation, ligand database was constructed. Ligand database was prepared in SDF format, which was converted to PDBQT format using cygwin32 commands. For docking purposes, a grid box with 0.5 Å spacing was generated using the dimensions of co-crystal ligand (X; −4.2437, Y; 4.8282; Z; 11.1699). In order to improve the docking accuracy, a total of 100 poses were generated with exhaustiveness value of 8. The best poses were selected on the basis of docking score and RMSD value and subjected to further analysis (Appendix A).

### 3.8. Molecular Dynamic Simulations

Nanoscale molecular dynamic (NAMD) [50] was used to conduct molecular dynamics simulations, while Visual Molecular Dynamic 1.9.3 (VMD) [51] was used to visualize the output trajectory. Initial protein ligand complex was collected from molecular docking studies, which was processed using CHARMM-GUI solution builder [52]. The rectangular-shaped water box extended 15 angstroms in each X, Y, and Z dimensions were used to solvate the system. The entire system was immersed in a TIP3P solvent model that was neutralized by the addition of 0.15M NaCl ions. Counter-ions were added using the Monte Carlo approach maintaining distance of 5 angstroms between each ion. The topology files of the system were generated using the CHARMM36 forcefield. The forcefield parameterization of compounds **6a**, **6e**, and **6g** was carried out using the ligand reader and modeler utilities of the online server CHARMM-GUI. Initially, ligand coordinates were saved in SDF format, which were processed using a ligand reader and modeler utility. After creating the necessary topology files, the minimization, NVT, and NPT ensembles were carried out using the same methods as described in our prior studies [53]. The created protein–ligand system was energy-minimized for 50,000 steps in order to eliminate any unwanted clashes, followed by equilibration in the NVT ensemble for 1 ns. Moreover, equilibration in the NPT ensemble was performed at a constant pressure of 1 bar and 300 K temperature for an additional 1 ns using Langvin thermostat. After equilibrating the system, the production run was performed for 100 ns and was conducted under periodic boundary conditions [54]. RMSD, RMSF, and SASA, among other analytic matrices, were utilized to evaluate the stability of protein–ligand complexes.

### 3.9. In Silico ADMET Prediction

In order to assess the absorption, distribution, metabolism, elimination, and toxicity (ADMET) profile of potent derivatives, in silico ADMET prediction studies were performed. These studies are important in characterizing the toxicity profile and drug-likeness of the compound [53]. In silico ADMET predictions substantially reduced the time of drug development in cost effective manner. In the current study, advanced deep learning models were used predict the ADMET profile of potent derivatives [35]. Specifically, the message-passing neural network model (MPNN) was deployed to assess the ADMET profile. Deep learning architecture took drug SMILES [55] as an input and provided precise predictions.

## 4. Conclusions

The development of 3-ethylaniline hybrid imino-thiazolines provided promising approach to target carbonic anhydrase II. Thus, in this study, a panel of ten 3-ethylaniline hybrid imino-thiazolines were synthesized and evaluated for their carbonic anhydrase inhibition activities. Among all these synthesized derivatives, **6e** potentially inhibited CA-II. Apart from an in vitro study, in silico studies including DFTs were performed, which showed that compound **6e** displays soft nature and potentially reacts with substrate protein CA(II). In addition, molecular docking studies demonstrated that compound **6e** possessed potential binding interaction with in active pocket of CA-II. In addition, pharmacophore modelling helped with figuring out the potential inhibitor of CA-II, as compound **6e** produced the best pharmacophore model with a dock score of −6.94 kcal/mol. A total of seven pharmacophoric features contributed to establishing the supramolecular interactions with a biological target. The stability of protein–ligand complex was ensured with MD simulations. Collectively, in vitro and in silico findings were suggestive of strong inhibitory potential of compound **6e** against CA-II.

## Data Availability

Not applicable.

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
