# Peer review of "Design, Synthesis, Kinetic Analysis and Pharmacophore-Directed Discovery of 3-Ethylaniline Hybrid Imino-Thiazolidinone as Potential Inhibitor of Carbonic Anhydrase II: An Emerging Biological Target for Treatment of Cancer"

_biomolecules, 2022, doi:10.3390/biom12111696_

Round 1

Reviewer 1 Report

Heterocyclic compounds play an important role in modern chemistry, especially in drug design and development. From this point of view, the present research focusing on the search for novel carbonic anhydrase II inhibitors among thiazolidinone derivatives is of certain interest. At the same time, some comments should be made regarding the research.

1. The graphical abstract appears to be informatively overloaded.

2. The hybrid term is more applicable to heterocycles linked via C-C, C-N, C-S, or other single bond types.

3. No clear explanation was given for why the 3-ethylphenyl moiety was selected as a substituent at the N(3) atom of the heterocycle.

4. In contrast to the introduction, which represents the synthesis of a novel ethyl (Z)-2-((Z)-3-((3s,5s,7s)-adamantan-1-yl)-2-((2-methylbenzoyl)imino)-4-oxothiazolidin-5-ylidene) acetate (6f) and its derivatives (6a-j), compounds of a different nature bearing the same numbers as those above, have been depicted in Figure 2.

5. In the abstract, mass spectrometry was pointed out as a chemical structure-confirming method. However, in section 2, whether molecular ion peaks were observed in the spectra or not, is not commented upon.

6. A successful structure-activity relationship analysis means excluding a random selection of substituents. Therefore, it should probably be explained what their selection was determined by. 

Author Response

Heterocyclic compounds play an important role in modern chemistry, especially in drug design and development. From this point of view, the present research focusing on the search for novel carbonic anhydrase II inhibitors among thiazolidinone derivatives is of certain interest. At the same time, some comments should be made regarding the research.

  1. The graphical abstract appears to be informatively overloaded.

Response: The graphical Abstract has been modified as suggested.

  1. The hybrid term is more applicable to heterocycles linked via C-C, C-N, C-S, or other single bond types.

Response: Yes, the reviewer is right here the Iminothiozolidinoe is linked with 3-ethylphenyl moiety through C-N bond

  1. No clear explanation was given for why the 3-ethylphenyl moiety was selected as a substituent at the N(3) atom of the heterocycle.
  2. Response: 3-ethylphenyl was selected as lipophilic tuning part of pharmochophore and to develop hydrophobic interactions of ligand with protein active site.

  1. In contrast to the introduction, which represents the synthesis of a novel ethyl (Z)-2-((Z)-3-((3s,5s,7s)-adamantan-1-yl)-2-((2-methylbenzoyl)imino)-4-oxothiazolidin-5-ylidene) acetate (6f) and its derivatives (6a-j), compounds of a different nature bearing the same numbers as those above, have been depicted in Figure 2.

Response: We named the first synthesized derivative as 6f and substitution was carried out variation of substituent on phenyl part of acyl moiety and name rest of them according to 6a-j. All the derivatives are in sequence.

  1. In the abstract, mass spectrometry was pointed out as a chemical structure-confirming method. However, in section 2, whether molecular ion peaks were observed in the spectra or not, is not commented upon.

Response: Sorry for the inconvenience, the data has been incorporated in the characterization part as suggested.

  1. A successful structure-activity relationship analysis means excluding a random selection of substituents. Therefore, it should probably be explained what their selection was determined by.

Response: Actually, the electron-donating substituents were selected in this work and corrections have been incorporated in the revised manuscript as suggested

Reviewer 2 Report

The authors have written "Design, synthesis, kinetic analysis and pharmacophore directed discovery of 3-Ethylaniline hybrid iminothiozolidinone as potential inhibitor of carbonic anhydrase II: an emerging biological target for treatment of cancer" well. The introduction and results discussion are well . The paper can be accepted with explaination of following points.

1) The difference in Electrophilicity index  values given in Table-2 and Table -3 explaination is not  given.

2) The Experimental part in main manuscript should be little bit more elaborated. Like line 348/349 whether it is just treatedb  or stirred  is not mentioned. In main manuscript.

Author Response

The authors have written "Design, synthesis, kinetic analysis and pharmacophore directed discovery of 3-Ethylaniline hybrid iminothiozolidinone as potential inhibitor of carbonic anhydrase II: an emerging biological target for treatment of cancer" well. The introduction and results discussion are well . The paper can be accepted with explaination of following points.

  • The difference in Electrophilicity index  values given in Table-2 and Table -3 explaination is not  given.

Response: Electrophilicity index values are revised in table 2 and omitted from table 3. It was typo error which has been omitted. In addition, explanation has been incorporated in the manuscript.

  • The Experimental part in main manuscript should be little bit more elaborated. Like line 348/349 whether it is just treatedb  or stirred  is not mentioned. In main manuscript.

Response: In experimental section, same methodology has been opted as reported in our previous studies (references have been added). However, as per suggestions, correction has been made. Experimental part including MD simulations and molecular docking have been revised.

Author Response

1.In line 143, authors mentioned acetazolamide as a reference compound to compare the inhibitory actions of derivatives. It would be great if authors show the chemical structure of acetazolamide and discuss its chemical structure in relation with the compounds investigated in this report.

Response: The chemical structure of standard inhibitor acetazolamide has been incorporated in figure 2 and detailed discussion of chemical structure in relation with synthesized derivatives (6a-j) have been provided in section 2.2. of main text.

2) The figure legends are short and it is hard for readers to interpret the meaning and the context of the figure as there is no sufficient explanation in the legend. Please elaborate the figure legends.

Response: All figures legends are revised with comprehensive explanation.

3) The study reports that 6a, 6e and 6g as top hits. However, the authors performed MD simulation for only 6a compound. MD simulations are generally used to assess the stability and binding energy of top hits. Therefore, I would ask the authors to consider the simulations for the 6a and 6g compounds too.

Response: Thankyou reviewer for substantial revisions. The compound 6e was top hit followed by 6a and 6g. Considering the suggestion of anonymous reviewer, the MD simulations for compounds 6a, 6e and 6g have been performed for 100 ns and incorporated in the main text as per suggestions.

4) RMSD doesn’t provide much intuition about the stability of the ligand as it is a global measure of protein stability. Instead, I would suggest analyzing the RMSD of ligands with respect to docking pose along the MD trajectory to assess the stability of ligands in the docked pose.

Response: The RMSD analysis of compound 6a, 6e and 6g was conducted in order to characterize the stability patterns as suggested. The changes have been incorporated in the main text.

5) I would suggest the authors to perform cluster analysis of 6a, 6e and 6g compounds using MD trajectory. Cluster analysis provides better evidence to assess the convergence or dispersion of docking pose in the binding site. Please refer to the following paper:https://pubs.acs.org/doi/10.1021/acs.jcim.7b00588

Response: Reviewer has suggested an impactful study encompassing multiple docking approaches, pose clustering followed by MD simulation to eliminate false positive. We greatly appreciate pose clustering technique as it facilitate in elimination of unlikely poses. In our current study, total 100 poses were generated for each derivative which were screened on the basis of energy rankings. As MD simulations are expensive approach so we have selected top 3 compounds on the basis of energy rankings for MD simulations with a goal of optimizing the ligand and further eliminating the incorrect poses. Initially stability of all three compounds i.e., 6a, 6e and 6g was evaluated but as per suggestions of reviewer pose clustering technique was also implemented which helped us in further eliminating the unlikely poses. The pose clustering for compound 6a, 6e and 6g was performed in the light of referee comments and data has been incorporated in the manuscript. Citation has also made in the manuscript.

6) If the authors have MD trajectory for the top three candidates, it will be a good practice to perform MM-PBSA or MM-GBSA approach to estimate free energies for the three top hits and compare with docking results.

Response: The suggested data has been incorporated in main text under section 2.5.5.

7) The authors didn’t explain how they generated the force field parameters for 6e molecules in their molecular dynamics study.

Response: The forcefield parameters for selected compounds were generated through ligand reader and modeler utility of online server CHARMM-GUI. The discussion has been incorporated in main text (section 3.8).

The authors should explain in more detail in the molecular dynamics simulations method section about the size and shape of the simulation box and also simulation parameters used in the study. Reporting the simulation parameters will improve the reproducibility of results in the future.

Response: we have used same simulation parameters as discussed in our previous study. However, considering the suggestion of reviewer, detailed simulation methodology has been incorporated in method section.

9) The docking interactions shown in Table 4 and Figure 7 are not consistent. Please be consistent with the number of interactions shown in the table with respect to Figure 7.

Response:  Thankyou anonymous reviewer for deep check, it was typo error which has been corrected.

10) How does the change of CH3 to Cl group increase the hydrophobic interactions, i.e 6g compound show more hydrophobic interactions than the 6e compound?

Response: Compound 6g was producing one additional hydrophobic interaction than compound 6e. As all synthesized derivatives possessed identical pharmacophore except single substitution of alkyl or halogen group on benzene ring. As in case of compound 6e, benzene ring was substituted with methyl group which was involved in total 4 hydrophobic interactions. Similarly, compound 6g have substitution of Cl group on benzene ring and this part was also involved in 4 hydrophobic interactions including alkyl, pi alkyl and aromatic interactions.  However, an additional hydrophobic interaction was observed with ethyl substituted benzene ring present on oxothiazolidine moiety of compound 6g. The detailed description has been also incorporated in the manuscript.

Minor comments:

1) Please check the following line. Does this scheme describe the synthesis of 6f compound or 6g compound? Please modify accordingly. Line 119-120:

Synthesis of (Z)-ethyl 2-((Z)-2-benzamido-3-(4-ethylphenyl)-4-oxothiazolidin-5-ylidene) acetate (6g) and its derivative (6a-j)”

Response: This scheme was describing synthesis of compound 6f, correction has been made in manuscript.

2) Please check the following line and it is misleading. Authors considered only one alkyl group i.,e R=methyl at different positions as substitution for the derivatives, but not the different alkyl groups. Line 133-134: “ Different alkyl groups and halogen atoms were substituted at phenyl ring in newly synthesized 6a-6j derivatives”

Response: Reviewer have spotted precisely, we have considered only methyl and halogen group for substitution. The misleading line has been corrected in the light of referee comments.

3) Please check the following line. I didn’t see any compound substitution with the C2H5 group. Please also check that the authors mentioned halogen atoms as electron donating? Somewhere in the draft, for example in line 169, the authors mentioned halogen atoms as electron withdrawing groups. It is contradictory, so please be consistent in the draft. Line 145-146: “ All ten 145 derivatives possessed electron donating groups i.e., CH3, C2H5 and halogen atoms”

Response: All confusing statements have been omitted form the manuscript. C2H5 mentioning line has been omitted.

4) The following sentence is not clear. Does it mean the current version? Please modify the sentence Line 418: “The current has utilized AutoDock Vina [46] for prediction of potential interactions between protein and 418 selected compounds”

Response: It means “current study” which has been corrected in the manuscript.

5) Please add the docking grid size in the molecular docking section and also illustrate the docking grid with a figure in the supplementary section.

Response:  Docking Grid size dimensions are incorporated in the molecular docking section. Illustration of docking grid has been also provided in the supplementary file

Reviewer 4 Report

Dear All

The manuscript here describes the synthesis and biological evaluation of a series of small molecules as CA II inhbitors for potential cancer therapy. The topic is somehow interesting for a broad range of audience and the text is clearly organized. However I found these major concerns before acceptance:

1. The manuscript lacks of a clear and detectable compounds characterization, please add (HPLC, MS data and 1H-NMR)

2. Statystical analysis should be provided for computational work, r value and p.

3. Introduction should consider also recent findings on CAs and its involvement in other human diseases for a sake of completeness; I suggest to consider the following literature: "A novel library of saccharin and acesulfame derivatives as potent and selective inhibitors of carbonic anhydrase IX and XII isoforms", "Novel 1,3-thiazolidin-4-one derivatives as promising anti- Candida agents endowed with anti-oxidant and chelating properties".

4. English needs revision for typos and grammar.

Author Response

Comments for Author

The manuscript here describes the synthesis and biological evaluation of a series of small molecules as CA II inhbitors for potential cancer therapy. The topic is somehow interesting for a broad range of audience and the text is clearly organized. However I found these major concerns before acceptance:

  1. The manuscript lacks of a clear and detectable compounds characterization, please add (HPLC, MS data and 1H-NMR)

Response: The characterization part has been corrected and information (IR, NMR and MS ) has been incorporated as suggested

  1. Statystical analysis should be provided for computational work, r value and p.

Response: The statistical analysis has been incorporated as suggested

  1. Introduction should consider also recent findings on CAs and its involvement in other human diseases for a sake of completeness; I suggest to consider the following literature: "A novel library of saccharin and acesulfame derivatives as potent and selective inhibitors of carbonic anhydrase IX and XII isoforms", "Novel 1,3-thiazolidin-4-one derivatives as promising anti- Candida agents endowed with anti-oxidant and chelating properties".

Response: Incorporated as Suggested

  1. English needs revision for typos and grammar.

Response: corrected as suggested

Round 2

Reviewer 3 Report

The manuscript has serious flaws and inconsistencies in the revised manuscript. In my opinion,  this manuscript has serious flaws and inconsistencies and therefore, I vote to not accept the manuscript. The authors didn’t revise the comments clearly and added more inconsistencies in the manuscript.

  • In the previous review, I have raised one of the important inconsistencies in the results. The author failed to reply to it. How does change of methyl to chlorine group increase the hydrophobic interactions? Instead it should be reversed as addition of chloride decrease the hydrophobic interactions ?
  • Why do 6e compound have more binding energy than 6g compound despite 6g making more interactions than 6e compound?
  • How does the change of methyl to chlorine group still retain three hydrophobic interactions with Leu198, Val121 and Val143 as shown in the figure 7c.
  • In the figure 7b and 7c, authors are claiming to show the interactions of 6e and 6g compounds respectively. No, they are not. The compounds in 7b and 7c are the same as there is no substituted chlorine group in the 7c figure?
  • Interactions in figure 7a and Table 4 for 6a compound are not consistent. Figure 7a shows 6a compound has interactions with phe131 whereas in table 4 this interaction is missing.
  • Why does 6a compound have less binding energy than 6e and 6g despite it having more hydrophilic and hydrophobic interactions?
  • The results in Figure 8 and Figure 9 look quite contradictory. For example, authors showed in Figure 8 that 6a compound made protein stable. Then how does 6a compound has more motion than the other compounds? And vice-versa for 6g compound?
  • In line 354, authors mentioned that 6g compound showed substantial fluctuations which is order of 6 ang? As per the figure 9, it doesn’t look like that. It looks like an average of 1.2 ang.
  • Average RMSD values of ligands reported in text and figure 9 are not matching. This entire section reported in the text is not consistent with figure 9
  • What is the importance of RMSF and SAS figure of apo protein. How does it tell about the binding affinity of ligands?
  • In line 368, authors mentioned that Figure 9 shows rmsf profiles of two protein but in the figure 9, they showed the rmsf profile of only one protein
  • In figure 11, the red structure moved very much apart from the binding site, but authors claimed that the pose deviation is 3 ang rmsd? Is it real?
  • In figure 11, the orientations of protein in the right and left panel are not comparable and it is hard for the reader to interpret .
  • Why the electrostatic interactions (Ecolumb) reported in Table 5 are positive?

Author Response

  • In the previous review, I have raised one of the important inconsistencies in the results. The author failed to reply to it. How does change of methyl to chlorine group increase the hydrophobic interactions? Instead it should be reversed as addition of chloride decrease the hydrophobic interactions ?

Response: Molecular docking figure is revised, now it is consistent and discussion is provided in main text.

  • Why do 6e compound have more binding energy than 6g compound despite 6g making more interactions than 6e compound?

Response: 6e is producing 3 hydrogen bonds and more hydrophobic interactions. Molecular docking discussion is revised.

  • How does the change of methyl to chlorine group still retain three hydrophobic interactions with Leu198, Val121 and Val143 as shown in the figure 7c..

Response: Molecular docking discussion has been revised, and it is depicting the clear scenario.

  • In the figure 7b and 7c, authors are claiming to show the interactions of 6e and 6g compounds respectively. No, they are not. The compounds in 7b and 7c are the same as there is no substituted chlorine group in the 7c figure?

Response: Figure has been revised.

  • Interactions in figure 7a and Table 4 for 6a compound are not consistent. Figure 7a shows 6a compound has interactions with phe131 whereas in table 4 this interaction is missing.

Response: Interactions in table 4 and figure 7 has been revised as per suggestions.

  • Why does 6a compound have less binding energy than 6e and 6g despite it having more hydrophilic and hydrophobic interactions?

Response: Compound 6e and 6a has structural similarity as they differ only in position of methyl group on benzene ring. In addition, both compounds were producing same interactions in terms of hydrophilic and hydrophobic interactions. However, compound 6e bonding strength was slightly better than compound 6a which might result in improvement of docking score.

  • The results in Figure 8 and Figure 9 look quite contradictory. For example, authors showed in Figure 8 that 6a compound made protein stable. Then how does 6a compound has more motion than the other compounds? And vice-versa for 6g compound?

Response: RMSD pattern for CA-II-6a complex remained stable throughout simulated trajectory. In terms of RMSD of compound 6a, initially it exhibited fluctuations which remained below 2.5 angstrom and after 40 ns, RMSD got stable and equilibrated below 2 angstroms. In contrast, compound 6g exhibited and more fluctuations and RMSD pattern of CA-II-6g was also slightly high.

  • In line 354, authors mentioned that 6g compound showed substantial fluctuations which is order of 6 ang? As per the figure 9, it doesn’t look like that. It looks like an average of 1.2 ang.

Response: It has been revised.

  • Average RMSD values of ligands reported in text and figure 9 are not matching. This entire section reported in the text is not consistent with figure 9

Response: RMSD average values are recalculated and revised in manuscript.

  • What is the importance of RMSF and SAS figure of apo protein. How does it tell about the binding affinity of ligands?

Response: RMSF values provide insight into residue wise fluctuation of targeted protein which is inevitable for determining the stability of protein. Whereas SASA values depicts the area of protein accessible by the surrounding solvent. Higher the value of SASA lower is the stability. But in current study, all residues depicted optimal SASA value.

  • In line 368, authors mentioned that Figure 9 shows rmsf profiles of two protein but in the figure 9, they showed the rmsf profile of only one protein

Response: It was typo mistake which has been ommited.

  • In figure 11, the red structure moved very much apart from the binding site, but authors claimed that the pose deviation is 3 ang rmsd? Is it real?

Response: The red colored pose deviation is 4.5 angstrom which is mentioned in the manuscript.

  • In figure 11, the orientations of protein in the right and left panel are not comparable and it is hard for the reader to interpret .

Response: Left side conformation is static which has been retrieved form molecular docking whereas right side conformation is dynamic state obtained from MD simulations.

  • Why coulomb energy is positive?

Response: Positive coulomb energy indicate the present of liked charged atoms (same charges) which might results in positive coulomb energy.

Reviewer 4 Report

Dear All

The manuscript has been revised according to referees suggestions, now it is suitable for publication.

Best Regards

Response: Thanks very much for your time and suggestions.